# Immunotherapies for the Treatment of Drug Addiction

**DOI:** 10.3390/vaccines10111778

**Published:** 2022-10-22

**Authors:** Md Kamal Hossain, Majid Davidson, Erica Kypreos, Jack Feehan, Joshua Alexander Muir, Kulmira Nurgali, Vasso Apostolopoulos

**Affiliations:** 1Institute for Health and Sport, Victoria University, Melbourne, VIC 3030, Australia; 2College of Health and Biomedicine, Victoria University, Melbourne, VIC 3021, Australia; 3Regenerative Medicine and Stem Cells Program, Australian Institute of Musculoskeletal Science (AIMSS), Melbourne, VIC 3021, Australia; 4Department of Medicine Western Health, Faculty of Medicine, Dentistry and Health Sciences, University of Melbourne, Melbourne, VIC 3021, Australia; 5Immunology Program, Australian Institute of Musculoskeletal Science (AIMSS), Melbourne, VIC 3021, Australia

**Keywords:** addiction, vaccine, substance abuse, immunotherapy, antigen delivery

## Abstract

Substance use disorders (SUD) are a serious public health concern globally. Existing treatment platforms suffer from a lack of effectiveness. The development of immunotherapies against these substances of abuse for both prophylactic and therapeutic use has gained tremendous importance as an alternative and/or supplementary to existing therapies. Significant development has been made in this area over the last few decades. Herein, we highlight the vaccine and other biologics development strategies, preclinical, clinical updates along with challenges and future directions. Articles were searched in PubMed, ClinicalTrial.gov, and google electronic databases relevant to development, preclinical, clinical trials of nicotine, cocaine, methamphetamine, and opioid vaccines. Various new emerging vaccine development strategies for SUD were also identified through this search and discussed. A good number of vaccine candidates demonstrated promising results in preclinical and clinical phases and support the concept of developing a vaccine for SUD. However, there have been no ultimate success as yet, and there remain some challenges with a massive push to take more candidates to clinical trials for further evaluation to break the bottleneck.

## 1. Introduction

Substance use disorders (SUDs) are a complex pathology of the central nervous system (CNS), representing a significant threat to health, social and economic wellbeing globally [1]. According to reports from the United Nations Office on Drugs and Crime (UNODC), in 2017, approximately 5.5 percent of the global population aged 15–64 (271 million people), used drugs in the past year [2]. Approximately 35 million people suffer from SUDs globally with the prevention and treatment of these issues continuing to fall short with only 1 in 7 people receiving appropriate treatment [2]. Collectively, smoking, alcohol, and illicit drug use kill 11.8 million people each year, more than all cancers combined [3]. In the US, the cost of substance abuse in lost productivity, and increased healthcare costs, as well as drug-related crime and its impact on the criminal justice system is estimated at $740 billion per year, and this figure is increasing [4]. When considered alongside other societal costs of addiction which are difficult to assign a monetary value, i.e., death from overdose, domestic abuse, unemployment, divorce, sexually transmitted diseases, homelessness, the total cost is likely to be considerably higher [4].

Managing SUDs and addiction is challenging and effective treatments are currently lacking. Existing pharmacological and psychological interventions have opened avenues to explore immunotherapies such as vaccines, monoclonal antibodies (mAbs), and catalytic enzymes, as the treatment platforms for the treatment of SUDs. Vaccines and mAbs receive particular attention due to their dual prophylactic and therapeutic applications [5,6]. Immunotherapies could be used as a complementary strategy to existing psychotherapy and pharmacological interventions to prevent addiction, relapse and overdose toxicity [7,8]. Beyond more traditional vaccine strategies, other approaches such as nanoparticle-based, and catalytically active vaccinations have recently gained attention from the scientific community as more data has emerged in these areas [9]. The development of immunotherapies against drugs of abuse is an established area, with a significant body of research underpinning these treatments. Herein, we describe all significant updates in the development of biologics, including vaccines, mAbs, drug catalyzing enzymes, and nano-vaccines to treat and prevent drug-induced addiction, including preclinical and clinical trials. We also highlight the challenges of the platforms and emphasize the role of regulatory bodies and funding agencies in expediting the progress of clinical translation. Finally, progress of emerging vaccine development strategies are presented.

## 2. Immunotherapies against Addiction: Mechanisms

Substances of abuse such as nicotine, heroin, cocaine, methamphetamine (METH), and morphine occur through similar mechanisms. After administration, the compounds rapidly perfuse into the CNS via circulation and affect reward processing, causing significant euphoric effects [10,11,12,13]. This euphoric response leads to repeated use, with the eventual development of addiction to the drug [14]. The repeated use of the drug is also reinforced by the negative physical and psychological effects of discontinuation. Anti-drug vaccines and mAbs work by a different mechanism than conventional pharmacological treatments. The immune system of the vaccinated individual will produce antibodies against the target compound in response to the administration of the drug. These bind to the drug in circulation, limiting their action on the CNS and other tissues. Drug overdose could be treated by the administration of mAbs resulting in rapid neutralization due to the drug-antibody complex being unable to cross the blood–brain barrier due to its size [10,11,12] (Figure 1). While vaccination strategies provide long-lasting effects, their antibody response is slower, meaning mAb treatment plays an important, albeit temporary, role in the reversal and treatment of overdoses due to their immediate action [13,14].

## 3. Platforms for Vaccine Development against Drug Abuse

Substances of abuse such as METH, cocaine, heroin, fentanyl, oxycodone, morphine, and nicotine work through similar mechanisms to induce addiction. Likewise, they all are small molecules, meaning they are too small to generate an immune response themselves. To achieve an immune response, they must be bound to an immunogenic carrier. Immunotherapies can be developed using various approaches. However, to date, vaccine development through hapten design and mAb by a process of genetic engineering or immunizing the genetically humanized mice are the most widely used strategies for developing immunotherapeutic interventions. The process of immunization can be categorized into active and passive strategies [15]. In an active vaccination, anti-drug vaccines are injected into patients to induce the host’s immune system to produce specific antibodies targeting the drug [16,17]. In passive vaccinations, a well-defined mAb is obtained through complex genetic hybridoma engineering either from animals or antibody libraries [9,13], with both approaches demonstrate promising results in animal models [10,11,12,18,19]. However, more recently, vaccine development approaches with hapten design and conjugation with an immunogenic carrier have received significant attention for their ability to modify the structure of compounds such as METH, cocaine, morphine, and nicotine, leading to the generation of a strong antibody response—a key criterion for successful vaccination [10,11,12]. The most common anti-drug vaccine and mAb development strategies are discussed below.

### 3.1. Vaccine through Hapten-Carrier Design

Conjugated vaccine development strategies through hapten design have been studied the most extensively for each of the substances of abuse [10,11,12,20,21]. The small size and chemical nature of substances of abuse render them unable to generate an immune response. To counter this, these molecules need to be conjugated to a known immunogenic carrier allowing for immune system recognition and effective processing [21,22,23]. This conjugated vaccine is fabricated by attaching a linker to the appropriate site of the drug molecule and this drug-linker complex (hapten) plays a critical role in the development of antibody and specificity [24]. Several factors can play a critical role in developing an anti-drug conjugated vaccine through hapten design and using protein carriers [25]. The substitution site of hapten, the type (alkyl/peptide) and length of the linkers, and the protein carrier type have a role to play. The developed haptens are attached to the protein surface and form a hapten cluster. This hapten density has a direct link to the efficient presentation of the antigen to the APC and subsequently antibody production. Bovine serum albumin (BSA), diphtheria toxoid (DT), tetanus toxoid (TT), and keyhole limpet hemocyanin (KLH) have been widely investigated as a protein carrier for antidrug vaccine development [26]. The development of antibody response using these protein carriers varied greatly based on the hapten density on its surface. Hence, careful consideration is required during the development of an anti-drug vaccine through hapten design. It is not only which hapten, protein carrier, or adjuvant are used [27], but also the dose and injection site can all influence the antibody production and the efficacy of the vaccine. All these critical factors need to be optimized through the design of experiment strategy (DOE) for the best outcomes of the anti-drug addiction vaccine development. Figure 2A shows a schematic diagram of anti-drug vaccine development through hapten design.

### 3.2. Monoclonal Antibody Development

In general, the antibodies for the treatment of drug addiction (passive immunization) are developed by immunizing animals with an immunogenic conjugate vaccine system [28]. The produced antibodies are then isolated and fully characterized, before being administered to patients to prevent the effects of drugs. However, more recently, the development of anti-drug mAbs involves the application of complex and time-consuming genetic engineering technologies [29]. Animals are immunized, and the antibody-producing B-cells are harvested from their spleen. These are fused with myeloma cells to form hybridoma which is then screened for appropriate antibody production, with the positive cells expanded in culture. The cells produce significant amounts of the desired antibody, which is harvested for therapeutic use [30]. The overall process of mAb development is represented in Figure 2B.

## 4. Anti-METH Immunotherapies

METH is a potent and highly addictive CNS stimulant and has an array of effects [31,32]. METH abuse is a growing concern to healthcare authorities globally, and as such, research efforts into vaccine and immunotherapeutic strategies are growing [10,11,12].

### 4.1. Active Immunizations

One of the first examples of a METH hapten “N-(4-aminobutyl) methamphetamine” conjugated vaccine using immunogenic bovine serum albumin (BSA) protein was reported in 1973 for the validation of a radioimmunoassay method [33]. The first anti-METH vaccine intended for human application was validated in rat models in 2001 [34]. A METH hapten was conjugated to keyhole limpet hemocyanin (KLH) for the assessment of effectiveness in a rat model. The vaccination generated antibodies against METH, both with and without subsequent exposure to the drug. Most earlier studies have emphasized hapten design, selectivity, and antibody production, however recently, the assessment of the efficacy of the developed vaccine using animal model has been emphasized [35]. Significant progress has been made in the area of anti-METH vaccine development and its efficacy assessment in the preclinical stage. A current update is presented in Table 1.

No METH vaccine related clinical study was found to assess the safety and efficacy in humans from a search in the ClinicalTrials.gov (accessed on 11 October 2022) registry. Despite some encouraging results from animal studies, no study has been translated to human clinical study. Considering the devastation caused by METH induced drug addiction, urgent measures are required to assess more METH vaccine candidates in clinical trials to facilitate a means for more data on the effectiveness of METH vaccines and the eventually approved vaccine for the treatment of METH induced drug addiction.

### 4.2. Passive Immunization with METH mAbs

Despite their complex development, mAbs have been investigated significantly to assess their effectiveness for the treatment of METH induced addiction in animal models. One of the major advantages of mAbs is that they can offer a quick onset of action compared to vaccines and hence can bind with the drug molecule in the blood circulation and prevention of its entrance into the brain. This feature is particulary important in treating overdose toxicity where quick action can be life saving. Most importantly, mAbs are very much drug specific and very minimal CNS side effects are expected [19,29,45]. A good number of preclinical studies have been conducted to investigate the various critical parameters including selectivity, affinity, and their effectiveness in attenuating METH induced behavioural effects. From these studies, it has been concluded that this mAb-based platform has the potential to treat overdose toxicity and a complementary treatment option to existing behavioural therapies. Despite very encouraging study outcomes from these preclinical studies, only one study has entered phase 1 and 2 human clinical trials using ch-mAb7F9. This clinical trial (NCT03336866) was designed to evaluate the safety, tolerability, effectiveness of the IXT-m200 (ch-mAb7F9) in healthy volunteers. The volunteers were challenged with four different METH dose and the distribution of METH to the brain was investigated. IXT-m200 demonstrated a significant increase in the METH AUC and Cmax, up to 30 and 8-fold respectively. Based on the favorable data from the previous study InterveXion is running another phase II (NCT04715230) trial to investigate the effectiveness of IXT-m200 inpatient with acute METH toxicity. The study is currently active and recruiting volunteers for the study. A summary of the updates of anti-METH mAbs in preclinical and clinical stages is shown in Table 1. However, hapten-conjugate vaccines, and anti-METH mAbs have been largely limited to preclinical investigation and require an investment of both effort and funding to take them to human trial and eventual translation.

## 5. Anti-Cocaine Immunotherapies

According to UNODC, there were 23 million cocaine users in 2018 worldwide [46]. Despite the effort of decades, there are currently no US FDA-approved pharmacological therapies for the treatment of cocaine addiction [47]. In the 1990s, catalytic antibodies were investigated for the treatment of cocaine addiction [48,49,50]. Catalytic antibodies bind to the target of interest, and hydrolyze the compound to render it as ineffective. In a rat model, catalytic antibodies prevented cocaine’s reinforcing and toxic effects [51,52]. In another study, catalytic antibodies were able to degrade cocaine in vitro [52]. Later on, researchers at The Scripps Research Institute, investigated catalytic antibodies [53] and compared them to conventional anti-cocaine vaccines, concluding that non-catalytic hapten designs were superior [54,55]. Since then, several anti-cocaine vaccines have been developed through hapten design and conjugation with a carrier protein such as KLH and BSA, demonstrating promising results in preclinical studies (Table 2). Cocaine catalyzing enzymes have been explored as an alternative and effective treatment platform to the existing platform already under investigation [56]. Cocaine hydrolase (CocH) enzymes have been investigated for effectiveness in animal models. Chen et al. reported the development of a novel CocH through fusing with an antibody fragment crystallizable (Fc). The newly developed CocH-Fc has a longer biological half-life compared to native CocH (107 h vs. 8 h) [57]. The newly constructed CocH-Fc increased cocaine metabolism even after 20 days of the first dose in rat models. Another study reported that butyrylcholinesterase (BChE) mutated into CocH can maintain enzyme concentration in the blood for an extended period and metabolize circulating cocaine [57,58]. Benzoic acid and ecgonine methyl ester are two common metabolites produced by cocaine and are known to have an impact on the heart and brain reward pathway. Mice treated with cocaine hydrolase gene transfer therapy did not have any negative impact on blood pressure and cocaine-induced locomotor activity even after a lethal dose of 80 mg/kg. Another study conducted by Collins et al. reported that double mutant cocaine esterase (DM CocE) was effective in preventing cocaine reenforcing, lethal dose toxicities, and convulsant effects in rhesus monkeys [59]. Collins et al. further investigated the effect of a repeat dose of DM CocE in rhesus monkeys [60]. In the control group, the plasma concentration of cocaine was significant after administration of a cocaine dose of 3 mg/kg. In the treatment group (0.32 mg/kg DM CocE), the cocaine was hydrolyzed rapidly and was below the detection limit within 5 to 8 min. Another study reported that gene transfer of CocE prevented the action of cocaine in the brain. FosB expression is an indicator of behavioral alteration in chronic drug abuse and an immunohistochemistry study showed that there was higher expression of FosB in the neostriatum in the control group compared to treatment group [61].

However, there have also been completed phase I and II human trials conducted using the anti-cocaine vaccine TA-CD. The TA-CD vaccine consists of succinyl norcocaine (SNC) conjugated to cholera toxin B [34]. In the phase I trial, TA-CD produced cocaine-specific antibodies in vaccinated participants at all dose levels (10 μg, 100 μg, and 1000 μg). However, peak antibody levels varied significantly between groups and declined significantly within three months and did not persist after 1 year [37]. In a phase IIa outpatient trial, TA-CD was given to 18 subjects at two dose levels, 100 μg for 4 injections or 400 μg for 5 injections over 3 months. A similar trend was observed in this study with variable antibody levels across the subjects, though mean levels were higher with larger doses. The high-dose group achieved and maintained higher levels of abstinence during the 3-month study period (71% vs. 44%). However, at 6 months, antibody levels had declined significantly, and relapses occurred in both low- and high-dose groups (89% vs. 43%). A phase IIb trial randomized 115 cocaine-dependent subjects in a methadone maintenance program to five injections of 360 μg TA-CD or placebo. Again, antibody levels varied significantly across the subjects with 38% achieving IgG levels ≥43 μg/mL but nearly one-third achieving less than 20 μg/mL, the level considered sufficient to block a single smoked cocaine dose based on previous studies [40]. Overall, actively immunized subjects showed no significant differences compared to placebo subjects in attaining complete abstinence.

To further investigate TA-CD vaccines, a large-scale phase II multicenter trial was conducted to compare the effect of five 400 μg doses of TA-CD to placebo on cocaine use over 8 weeks. Cocaine dependent participants were recruited from 6 centers across the USA. Initial vaccination was followed four boosters were given at weeks 3, 5, 9, and 13. High levels of anti-cocaine antibodies (≥42 μg/mL) were observed in 67% of the participants [39,46]. Vaccinated participants-maintained abstinence for at least 2 weeks of the trial after week 8 (24% vs. 18%), and the high-IgG group had more cocaine-free urine samples during the final 2 weeks of treatment. Nevertheless, again, there was no significant difference in cocaine-positive urine rates among the three groups (placebo, high-IgG, low-IgG) over the full course of the trial. The preclinical and clinical studies of anti-cocaine vaccines, mAbs, and catalytic antibodies are presented in Table 2.

## 6. Anti-Nicotine Immunotherapies

Smoking tobacco is one of the most significant preventable causes of death globally [14], and is responsible for approximately 6 million annual deaths [68]. The major addictive compound present in tobacco is the alkaloid nicotine, which is rapidly absorbed from the smoke and localises to the brain, where it binds to nicotinic cholinergic receptors, resulting in the release of dopamine in the CNS [69]. Dopamine release leads and the subsequent dysregulation of reward behaviors lead to the initiation of addiction. Pharmacological avenues for smoking cessation are limited, with only nicotine replacements (patches, or gums), the antidepressant bupropion, and the nicotinic receptor agonist varenicline, all of which seek to support tobacco smokers in managing their cravings while they reduce their smoking [70]. Despite these being available, less than a third of smokers seek medical assistance in quitting, and only a third of those successfully cease smoking for more than 6 months [71,72]. Additionally, more than 50% of individuals relapse following pharmacological treatment for smoking cessation [73]. These pharmacological interventions are strengthened by the addition of psychological treatments, with combination approaches having twice the efficacy of either intervention alone [74]. In the face of these challenges in promoting the cessation of smoking, new approaches are needed to control this critical health issue. Immunotherapeutic interventions have shown promise in the management of tobacco use and may provide avenues for promoting long-lasting cessation of smoking (Table 3).

Vaccine strategies against nicotine were first reported in the 70’s, with the development of a conjugated hapten vaccine. A trans-3′-succinylmethylnicotine hapten was conjugated to KLH and applied to rabbits in order to generate antibodies with which to assess the levels of nicotine in the blood and urine of humans [78]. Shortly after, BSA conjugated vaccines using the 6-(p-aminobenzamido) nicotine hapten were developed and again shown to produce an antibody response against nicotine in animal models [79,80]. The antibodies produced were shown to have strong binding affinity for nicotine at several binding sites, making a strong case for their use [81,82]. However, active vaccination strategies are yet to be shown to be effective in modulating behaviour. Rats immunized with a conjugated vaccine consisting of a 6-(carboxymethylureido)-(±)-nicotine hapten conjugated to KLH produced antibodies following vaccination, which bound a significant amount of the nicotine in plasma, did not affect the amount of nicotine in the brain, limiting the likely effectiveness [83]. More modern approaches have had more success in animal experiments. A vaccine using a nor-nicotine hapten conjugated to KLH via a novel linker protein (coined NIC) [84] led to decreased levels of nicotine in the brain of rats, along with decreased motor activity in response to a nicotine challenge [85]. However, two conformationally constrained haptens, N-[6-(2,3,3a,4,5,9b-hexahydro-1H-pyrrolo [2,3-f]quinolin-1-yl)hexanoyl]-β-alanine (CNI) and N-[6-(2,3,3a,4,5,9b-hexahydro-1H-pyrrolo [3,2-h]isoquinolin-1-yl)hexanoyl]-βalanine (CAN), when conjugated to KLH, showed increased antibody production over the NIC-KLH [86,87].

A vaccine using the nicotine analogue rac 6-((trans-1-methyl-2-(pyridin3-yl)pyrrolidin-3-yl)methoxy)hexanoic acid (dubbed AM1) hapten, conjugated to adenovirus hexon protein led to the production of significant antibody titers, which lasted more than 20 weeks in mice. The AM1 vaccine also reduced the levels of nicotine in the brain and suppressed locomotor activity following nicotine administration [88]. These findings of hapten superiority over one another likely stem from the significant alterations in immunogenicity caused by even small modifications to conformation. Indeed, linker length and flexibility seem to play a significant role, with longer and more mobile proteins, joined at the 6 position of the pyridine ring gave the strongest immune response [89]. Immunogenicity can also be improved through adjuvant or nanoparticle delivery systems, leading to improved immune responses [90,91]. A recent nanoparticle vaccine using a poly(lactide-co-glycolide) acid (PLGA) and KLH core, surrounded by a lipid bilayer studded with rac-trans 3′-aminomethyl nicotine haptens, showed up to a 400% increase in antibody production compared to protein conjugate vaccines [92]. This nanoparticle vaccine was shown to be even more effective with the addition of a Toll-like receptor (TLR) adjuvant which stimulates innate immune recognition and uptake, further decreasing nicotine levels in the brain [93]. Another nanoparticle vaccine using negatively charged, carbon nano-horns and O-succinyl-3′-hydroxmethyl-(±) nicotine hapten enclosed in liposome nanoparticles also showed superior immune reactivity to BSA-hapten conjugate vaccines [94]. While these preclinical vaccine models are promising, human trials are yet to demonstrate strong clinically relevant outcomes.

Several vaccine candidates have entered human trials, however broadly, results are discouraging. The 3′-AmNic-rEPA vaccine or ‘NicVax’ is a 3’-aminomethylnicotine hapten conjugated to pseudomonas aeruginosa exoprotein A, showed beneficial responses in a phase I trial of 68 smokers, with higher doses leading to higher levels of abstinence, and the treatment having a favorable safety profile [95]. Phase II trials also showed positive results, with strong antibody results and increased smoking abstinence in the treatment group compared to placebo [96]. The vaccine entered phase III trials, however failed to reach its endpoints, both as a single intervention and in combination with varenicline and counseling [97]. A tetanus-toxoid conjugated hapten vaccine called ‘Niccine’ was shown to have similar effects in trials, with an acceptable safety profile, but no change to smoking behaviour [98]. Another candidate vaccine named NIC-002, in which nicotine is conjugated to a virus-like particle, also failed to meet clinically significant endpoints in trials, showing a small, but statistically significant increase in abstinence at 2 months, but no difference after 6-months [99]. Another candidate, TA-NIC, failed to meet its clinical endpoints and was discontinued. A nanoparticle-based intervention, designated SEL-068 has been completed phase I human trials after showing efficacy in animal models. SEL-086 is a biocompatible polymer nanoparticle, with a synthetic TLR agonist, a novel T helper cell peptide, and nicotine covalently bound to its surface. This vaccine has shown strong results in mice and non-human primates, generating strong antibody titers with a high affinity for nicotine [100]. However, no results have been posted in the public domain from the phase 1 clinical trial. Pfizer conducted a phase I clinical trial to investigate the safety and tolerability of multiple doses of NIC7-CRM vaccine in healthy volunteers. The was completed in 2015 and no results have been reported. The NIC7 is a conjugated vaccine with alum and CpG as adjuvants to increase antibody production [101]. A preclinical study conducted using this formulation demonstrated an 80% reduction of nicotine in the brain and supported the conduct of a human trial [101].

Nekhayeva et al. developed a nicotinic monoclonal antibody named Nic311 and assessed the effectiveness of the mAb to prevent the transfer of drugs from maternal to fetal circulation [102]. The authors also claimed that there was more protein-bound drug and less free drug in the serum which suggests that this mAb is effective in the prevention of drug migration from the maternal to the fetal circuit. Another study conducted using the Nic311 mAb investigated its effectiveness in the acute and chronic distribution of nicotine [103]. The authors reported that Nic311 reduced the early distribution of nicotine to the brain but could not reduce the chronic accumulation of nicotine in the brain. Another study also reported that Nic311 was effective to reduce the Nicotine level in the rat brain [104]. Nicotine concentration was reduced by 45%, 83%, and 92% when Nic311 was administered at 30, 80, and 240 mg/kg body weight respectively. These doses also attenuated nicotine induced behavior alteration. A very recent study reported the isolation of a high-affinity mAb called AT-1031 from the B cells of smokers [105]. They investigated the effectiveness of AT-1031 in rat models by administering doses of 40 and 80 mg/kg and the nicotine concentration in the rat brain was reduced by 56% and 95% respectively.

Several studies have also investigated the efficacy of drug catalyzing enzymes to prevent nicotine addiction. Pentel et al. reported that NicA2 nicotine-degrading enzyme isolated from *P. putida* was effective in reducing nicotine concentrations in the blood by more than 90% within 1 min of nicotine administration [75]. Nicotine levels in the brain were significantly lower than in the control group. A NicA2 dose of 70 mg/kg reduced nicotine self-administration. Another study reported that a chronic dose of reengineered nicotine catalyzing enzyme (NicA2) completely prevented the entry of nicotine to the brain, reduces nicotine like compulsive behavior in an animal model [76]. Another study conducted by Thisted et al., reported that a new variant namely A107R reduced the distribution of nicotine in rats’ brain 3-fold more than a wild type variant. This study also reported that PEGylation of the enzymes improved the circulation half-life [77].

Despite ongoing work in the area, there is yet to be an effective agent introduced into human trials. Second generation vaccination approaches such as nanoparticles and peptide-based immunotherapies may provide the innovation needed to generate clinically meaningful outcomes, however, more research into agents, haptens, and adjuvants is required to identify potential candidates.

## 7. Anti-Opioid Immunotherapies

From 1999 to 2019, nearly 500,000 people died from an overdose involving any opioid, including prescription and illicit opioids [106]. Opiate addiction is a significant concern and efforts to develop successful treatment options have always been a priority. The development of an effective heroin vaccine began in 1970 with the identification of an immunogenic morphine hapten for use in radioimmunoassay [107]. This become particularly relevant in the development of a heroin vaccine shortly after, when it was reported that antibodies against morphine were identified in heroin users [108]. In what is considered the first anti-drug vaccine trial, drug-seeking behaviour in a rhesus monkey after the administration of a novel heroin vaccine was shown to be significantly decreased [17] and later a delay in morphine clearance in rabbits immunised with morphine-6-hemisuccinate-BSA as a result of morphine-antibody binding was described [109]. Importantly, because heroin is hydrolyzed to 6-acetylmorphine in the serum, vaccines against morphine will also immunize against heroin [35]. Despite early successes, immunotherapies against opiate addiction were abandoned at this time, likely due to the availability of other pharmacotherapies such as methadone and naltrexone [110]. Years later, a morphine-6-hemisuccinate-BSA hapten vaccine was tested in mice; generating anti-morphine antibodies 8- weeks post-vaccination [111]. Shortly after, a similar vaccine utilizing 6-succinylmorphine conjugated to BSA was tested in BALB/c mice, producing similar results [112]. Ten years later, the same vaccine was shown to be able to generate and sustain anti-morphine antibodies in the serum of humans over one year [113]. Another vaccine using a morphine-6-hemisuccinyl hapten is covalently coupled to a tetanus-toxoid protein, instead of the more well described BSA carriers [114]. This design allowed a larger distance between hapten and carrier protein and the increased physical space increased the immunogenicity of the hapten as an antigen, by increasing the number of binding sites available [115]. This vaccine achieved and maintained very high anti-heroin and anti-morphine antibody titers in mouse models. A well regarded research team at Walter Reed Army Institute of Research led by Dr. Gary Matyas developed and investigated a number of conjugated vaccines in animal model [115]. One of their candidates was developed by conjugating MorHap with tetanus toxoid (TT) using a PEGYlated linker. The formulation was incorporated into a liposome containing monophosphoryl lipid A and this formulation is commonly known as Army Liposome Formulation (ALF) [116,117,118]. This formulation MorHap-TT/ALF demonstrated strong antibody response in mice and animal models and reduced the motor activity and tail flick analgesia. This formulation demonstrated more efficacy when ALF was adsorbed on alum [119]. The efficacy of MorHap-TT/ALF was evaluated by coadministration with a HIV vaccine and the effectiveness of the MorHap-TT vaccine was not affected or reduced [116].

Another vaccine using a novel 6-glutarylmorphine hapten conjugated to KLH showed strong anti-morphine/heroin antibody responses and attenuation of drug seeking behavior in mouse models [120]. The rapid metabolization of heroin into several other psychoactive substances is an obstacle to immunization, however, the development of a dynamic hapten has allowed this to be countered [121]. A vaccine was developed in which heroin was coupled to BSA and KLH at the bridge nitrogen atom with alum used as an adjuvant. Degradation of the hapten at C3 and C6 following immunization allowed for immunity against the degradation products of heroin. Vaccine immunogenicity testing in rats elicited a robust immune response with a strong affinity to 6-acetylmorphine and a good affinity to morphine and heroin.

The last ten years have seen a considerable increase in focus on vaccination against drug abuse. A decrease in brain morphine levels after vaccination with a KLH-6-succinylmorphine hapten vaccine was shown in rats [122]. Additionally, increased anti-heroin antibody titers were shown in rats which were vaccinated intraperitoneally and subcutaneously vs. subcutaneously alone, as well as in rats who received the TLR9 agonist cytosine-guanine oligodeoxynucleotide 1826 (CpG ODN 1826) in addition to an alum adjuvant, suggesting that adjuvants which prime the innate immune response could be used to generate more effective responses [123]. Another heroin conjugate vaccine utilizing tetanus-toxoid with alum and CpG oligodeoxynucleotide adjuvants was developed based on this work. This novel vaccine was shown to reduce heroin potency by more than 15 times and produce lasting effects; serum antibodies were increased even 8 months after the initial vaccination [124]. Another large study compared 20 different anti-heroin vaccines with varying combinations of conjugates and adjuvants. Two adjuvants were tested for their added immunogenic effects and storage stability: the TLR9 agonist CpG ODN 1826, and the TLR3 agonist, virus-derived genomic double-stranded RNA (dsRNA). Both were found to elicit a strong anti-heroin immune response, however, only CpG ODN 1826 was stable after one month in storage. Interestingly, the combination of the two adjuvants did not elicit a stronger immune response [125]. Another study also looked to improve efficacy and address the short shelf-life of stored vaccines through the exploration of different adjuvants and storage conditions. It showed that inulin based and CpG ODN containing adjuvants combined with heroin hapten conjugates produced a robust immune response in mice, and that freeze-drying was an effective storage method for opioid vaccines; efficacy of these vaccine formulations was maintained for up to 1 year at room temperature [126].

Fentanyl is another member of opioid group and is like morphine. It works by binding with opioids receptor in the brain and can reduce pain sensation and enhance the pleasure effect [127]. Fentanyl is heavily abused to adulterate other opioids and is a serious concern globally. Fentanyl is accountable for about 30,000 deaths out of 50,000 total deaths caused by opioids in the USA [128]. Efforts are underway to develop immunotherapies to treat/prevent fentanyl addiction, and adverse outcomes. As with other substances of abuse, fentanyl is non immunogenic by itself and must be conjugated with an immunogenic carrier. Several studies have been conducted to assess the safety and efficacy of fentanyl conjugated vaccines in preclinical and clinical stages. In addition, fentanyl conjugated vaccine via intranasal, intramuscular and sublingual routes, formulated using three adjuvants—alum, and the *E. coli* derived dmLT and LTA1 were assessed [127]. It was noted that fentanyl vaccines containing dmLT or LTA1 adjuvants produced a strong antibody response, and protected penetration of fentanyl to the brain when administered sublingually. In another study anti-Fentanyl mAbs reduced fentanyl induced bradycardia and respiratory depression which are the major risks of fentanyl related fatality [129]. A fentanyl conjugated vaccine using a KLH carrier was successful in preventing the distribution of fentanyl in the brain of mice and rats [130]. Further, fentanyl–tetanus toxoid conjugated vaccine in a rat model [131], was successful in preventing fentanyl reinforcement, and increased the food uptake over the fentanyl administration [131]. However, conjugated fentanyl vaccine incorporated into a liposome formulation with monophosphoryl lipid A produced a strong antibody titer. The fentanyl antinociceptive dose–response curve was also been shifted to a higher dose level [132].

Oxycodone (OXY) is another member of the opioid family, used widely as a strong painkiller and is widely abused worldwide. Existing pharmacological treatments are somewhat effective but are associated with many side effects [133], making the development of immunotherapies as an alternative treatment platform of significant importance. A number of preclinical studies have been conducted to assess the effectiveness of vaccines with one candidate translated into human clinical trials. The OXY-(Gly)_4_-BSA/KLH vaccine was investigated in a rat model, demonstrating strong antibody titers and was effective in reducing hotplate analgesia and maintaining acceptable serum OXY levels. KLH conjugated OXY vaccine (OXY-KLH) in mice [133], showed that the vaccine attenuated the oxycodone reinforcement to clinically acceptable levels. Another OXY vaccine incorporated a glycine linker at the C6 position of OXY and conjugated to KLH (6OXY-(Gly)_4_-KLH). Both OXY and hydrocodone based vaccines were investigated in rats and mice and 6OXY-(Gly)_4_-KLH vaccine was effective in serum drug binding, reducing drug distribution to the brain and reducing analgesia [134]. Another study used the OXY-dKLH vaccine in mice to assess safety and efficacy [135]. The vaccination shifted the oxycodone dose–response curve to a higher dose, reducing bradycardia and respiratory depression [135]. A phase I/II (NCT04458545) study has been registered to investigate the effects of oxy(Gly)_4_-sKLH vaccine against oxycodone. This is a multisite study and is currently actively recruiting volunteers to assess the safety, degree of antibody, and efficacy.

To date, no heroin or morphine vaccines have been approved for use in humans. Additionally, no human trials, past or present, have been registered. However, the past 50 years have seen progress from proof of concept to efficient and efficacious vaccine design in preclinical models. Evidence continues to support the notion that vaccination may be a powerful tool in the fight against opioid addiction, however, research effort is needed to push these concepts into human trials. A snapshot of the vaccine and mAbs effort against opioids has been presented in Table 4.

## 8. Biomarkers and Vaccine Efficacy

Antibody response has been considered a first line indicator of immunogenicity and efficacy in most vaccine candidates [136] and the same concept is applicable to vaccines against SUDs. Many vaccine candidates against SUDs have demonstrated strong antibody responses in animal models but failed to do the same in human clinical trials. Significant variation in antibody responses; however, amongst human participants has been noted [137]. To overcome this hurdle and increase vaccine development and clinical trial success, a biomarker analysis as an indicator of vaccine efficacy has been explored. The correlation between 6 OXY-KLH and 8 HYDROC-KLH hapten mediated naïve B cell activation and vaccine efficacy in mice model has been evaluated as a proxy marker. There are millions of naïve B cells in all mammals of which a small portion become activated and capable of binding to the vaccine/hapten and initiating the complex process of proliferation and differentiation which eventually produce antibodies. Taylor et al. showed that naïve B cells showed greater affinity to 6 OXY before immunization and that these 6 OXY specific naïve cells had a greater affinity for free oxycodone. Once mice were injected, the 6 OXY-specific B cells were detected before the antibodies, suggesting earlier evidence of vaccine efficacy or failure. Another strategy revolves around IL-4, a key immune checkpoint inhibitor, with the hypothesis that blocking or depletion of IL-4 could increase vaccine efficacy, with reports that OXY-KLH vaccine efficacy increased in the absence of IL-4 [138]. However, IL-4 could also be a potential biomarker to determine vaccine efficacy for SUDs [139]. Blocking of type 1 IL-4 increased the vaccine efficacy for oxycodone and fentanyl and the absence of IL-4 receptor did not increase the efficacy against oxycodone and fentanyl. The experts believe that the assessment of biomarkers such as naïve and activated B cells, and IL-4 could be explored more as a biomarker of immunotherapies efficacy against SUDs which will ultimately increase vaccine development success and reduce the chances of clinical trial failure [138,140,141].

## 9. Expert Opinion

The history of vaccines and other immunotherapies for the treatment of substance abuse disorders is 5 decades old. There has been significant progress over the last two decades in the development of vaccines and other immunotherapies, with some of the platforms being investigated in clinical trials. For example, vaccines against nicotine and cocaine as well as anti-METH mAbs have been investigated in clinical trials, however, these have mostly failed due to a lack of and large variations in antibody response in the participants. Although no ultimate success has been achieved, these studies have provided critical information based on which scientists are working to improve the efficacy of the investigated platforms. Apart from conventional vaccine platforms, some others such as gene therapy, drug degrading enzymes, particle-based vaccine, nano-vaccine, and self-assembled nanofibers have been quite effective in other immunotherapeutic settings and could be explored for the development of a successful treatment option for substances of abuse. Despite significant progress over the 5 decades, several challenges have been identified and require attention to ensure the success of these platforms.

The saturation of antibodies by the drug or overdose of the drug is always a real concern. If the available antibodies are saturated by the overdose of the drug, there will be still free drugs for their addictive activities. There is a concern about the level of antibody production by the various vaccines and mAbs candidates in clinical trials. However, no benchmark on the optimal level of antibody responses has been set. It has been always thought that a very high antibody response is required for the success of the vaccine and mAbs platforms. This makes evaluation of clinical trial results (particularly in early phase trials) difficult and could lead to incorrect assessment of failure. More data on the clinical response of the individual are required to fully evaluate efficacy. People use drugs at different frequencies and volumes and adequate antibody responses are essential to fight back. However, most of the preclinical and clinical studies did not consider this factor in their studies. People may be vaccinated while they are in rehabilitative care, regularly using a drug or occasional using. In each situation, a different level of antibody response and treatment protocol and should be investigated by simulating these scenarios in preclinical and clinical studies.

Immunotherapies against SUDs are going to be complementary to existing behavioral treatments and studies are to be conducted on patients already in behavioral therapy to simulate the real-time scenario and to increase the success of the therapies. A combination approach like the administration of mAbs for quick action and vaccine for long-term effectiveness should be considered with drug addicts in the rehabilitation center. Drug abuse is a global issue with a serious impact at the national, community, and family levels, making the development of a successful intervention critical, and yet there is little research effort currently in this field. This is largely due to funding issues, and a lack of appropriate legislation to support these platforms once they are successful. Clinical trials are very expensive, and investors must consider a market to provide returns. However, research institutes and potential developers do not see a viable market due to this lack of appropriate legislation and do not feel encouraged to invest in this area. This is a global issue and funding should be allocated from government sources to engage more researchers to ensure translation of these immunotherapeutic approaches. Many of those living with SUDs use multiple drugs at a time. This means that a vaccine or immunotherapy for a particular drug may not give any significant benefit to polydrug users, and this area requires further investigation.

## 10. Novel Drugs of Abuse

The use of novel drugs for abuse is on the rise and is of great concern. The novel drugs of abuse are also known as designer drugs, legal highs, psychoactive substance and research chemicals [142,143]. The novel drugs of abuse are new synthetic compounds or a synthetic analogue of known substances of abuse. The commonly used novel substances of abuse are synthetic cannabinoids, sedatives (Phenibut, g-Hydroxybutyrate and associated compounds and kava), hallucinogenes (phencyclidine and analogues, lysergic acid diethylamide and associated analogues), opiate anlogues (fentanyl, analogues, and synthetic opioids and other serotonergic agonists), loperamide, and kratom [143,144]. Although there are no accurate epidemiological data on the use of novel drugs of abuse, it is increasing and can be associated with significant mortality, morbidity and other medical consequences [144]. The detection methods for these novel drugs of abuse have not been well established as the new drugs are continuously evolving which also makes it difficult to diagnose and take appropriate measures in case of medical consequences. As of now, no effort has been made to treat novel drugs of abuse. However, a preliminary study was conducted to assess the specificity of hallucinogenic compound lysergic acid diethylamide antibody specificity in rabbits and guinea pigs [145]. No other studies have reported on the effectiveness of the LSD antibody in treating LSD induced addiction.

## 11. Conclusions

Drug addiction is a serious neurological disorder and is quite often difficult to treat. Psychological therapy remains the sole treatment option. Biologics including vaccines, mAbs, drug catalyzing enzymes, and other platforms appear as promising options for the treatment of drug addiction. Significant development has been done over the last few decades on understanding the drug addiction mechanism, addiction-related disorders, the development of biologics, its effectiveness in preclinical and clinical trials, and have enriched this field tremendously. However, no ultimate success has been achieved and only a few candidates have been investigated in clinical trials. A few challenges have been identified and should be the focus of the research in the coming days to ensure the success of this platform. Suboptimal antibody response was the root cause of clinical trial failure for several candidates and in the coming days, the focus should be to increase the antibody response by using various strategies such as hapten design, adopting emerging platforms like nanocarrier vaccine, particle-based vaccines, and using the adjuvant system. Develop a user-specific protocol for the success of this platform and conduct more animal studies and clinical trials simulating this user-specific scenario. Funding and appropriate legislation should be ensured to attract potential developers, research institutes, and pharmaceutical companies to engage themselves for the greater benefit of this platform. Investigate the suitability of developing a cocktail vaccine or other immunotherapies to see the suitability of treating polydrug addiction.

## Figures and Tables

**Figure 1 vaccines-10-01778-f001:**
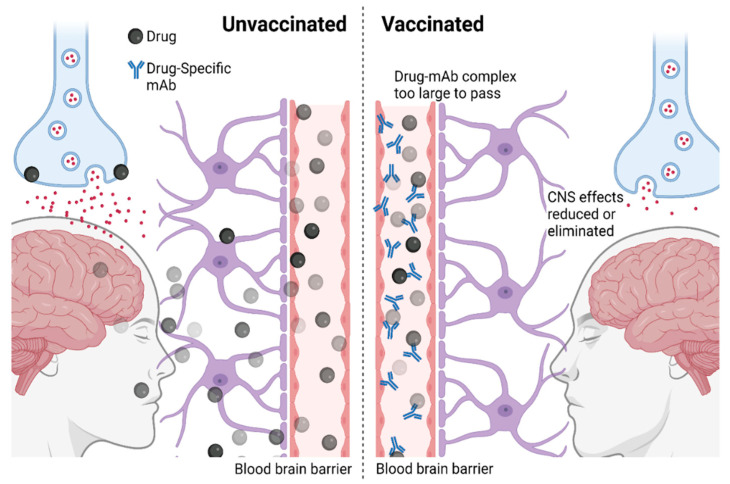
Mechanism of action of anti-drug immunotherapies. CNS: Central nervous system, mAb: monoclonal antibody.

**Figure 2 vaccines-10-01778-f002:**
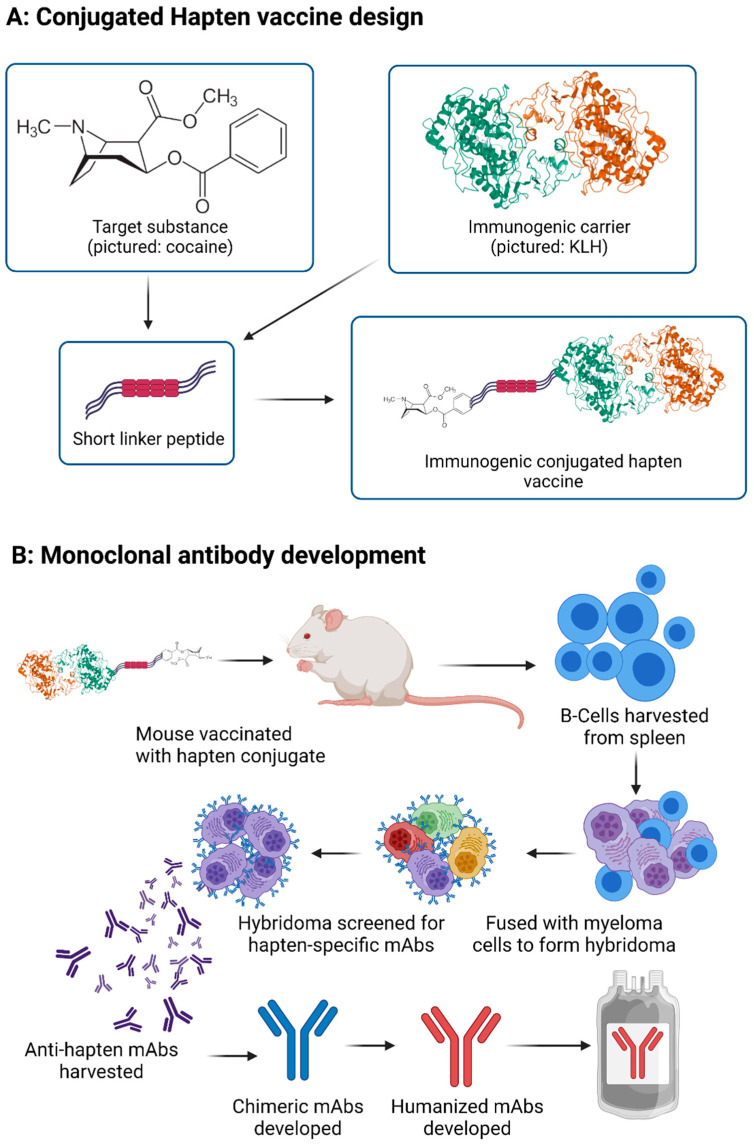
(**A**) Generation of conjugated hapten vaccines. (**B**) Generation of monoclonal antibody treatments. mAb: Monoclonal antibody, KLH: keyhole limpet hemocyanin.

**Table 1 vaccines-10-01778-t001:** Summary of preclinical and clinical studies using various anti-METH vaccines and mAbs.

Preclinical and Clinical Studies of Anti-METH Vaccines				
Hapten System	Carrier	Animal	Study Outcome	Ref
N-(4-aminobutyl) methamphetamine	BSA	Mice	Development and validation of a radioimmunoassay	[33]
(+) METH with a six-carbon spacer group at the para position of the ring structure	KLH	Sprague–Dawley rats	Rats injected with METH hapten-KLH conjugated vaccine developed METH antibody titers	[34]
MH1-MH7	KLH	GIX Mice	METH haptens MH2, MH6 and MH7 conjugated with carrier proteins demonstrated very promising anti-METH antibody titer and METH affinity in mice model	[36]
MH6	KLH	Rats	METH haptens MH6 conjugated with KLH reduced METH-induced thermoregulatory and locomotor effects	[37]
SMO9	KLH	Rats	SM09 conjugated with KLH prevented rats from METH-induced impairment of food responses	[38]
N-succinylmethamphetamine (SMA)	KLH and TT	Mice	SMA-KLH/TT conjugated vaccine reduced the METH-induced hyperlocomotion successfully in the mice model	[39]
Nine (9) METH haptens with alkylating and peptide linkers	TT and DT	Webster mice	Hapten 12 conjugated with TT & alum/CpG ODN 1826 as adjuvant demonstrated excellent antibody titer (300,000) and reduced METH-induced locomotor activity compared to controls	[21]
MH6	KLH	Wistar rats	MH6-KLH conjugated vaccine reduced the locomotor activity caused by 0.25 mg/kg METH administration by IP injection.	[40]
**Preclinical and Clinical Studies Using Anti-METH mAbs**				
**mAbs Description**	**Model Tested**		**Study Results**	**Ref**
Assessment of mAbs with different affinity	Rats		Two anti-METH mAbs with different affinities were assessed and mAb with higher affinity were more effective to reduce the locomotor activity	[34]
Anti-(+)METH mAbs (mAb; KD = 11 nM)	Sprague–Dawley rats		A significant reduction in METH concentration in brain (>60%) and increase in METH serum concentration (>6600%)	[19]
Murine derived anti-METH mAbs (mAbH4 & mAbH8)	Rats		Both of the mAbs were effective in retaining METH in the circulation and prevented METH entry in the CNS and its subsequent effects.	[41]
Assessment of anti-METH and anti-phencyclidine (PCP) mAbs combination	Pigeons		This combination demonstrated significant selectivity and prevented the drug induced behavioral effects.	[42]
Human-mouse chimeric mAb for (+) METH with high selectivity and affinity	Rats		ch-mAb7F9 mAb demonstrated excellent binding potential with METH and altered the distribution (leass METH in brain and more in blood) of METH and clearance from the body.	[43]
Chimeric -mAb7F9	Rats		ch-mAb7F9 demonstarted attenuation in addiction related effect after METH acute dose	[44]
ch-mAb7F9	Humans		A Phase 1 clinical trial including 42 volunteers. A dose of 0.2 to 20 mg/kg bodyweight was administered to assess the safety of ch-mAb7F9 in humans. No adverse effect were reported	NCT01603147
IXT-m200 (ch-mAb7F9)	Humans		A phase IIa study was conducted to investigate the effects of anti-METH mAb and their effectiveness to retain the METH in the bloodstream. This is a randomized interventional clinical trial of 126 participants. The study demonstrated that IXT-m200 (ch-mAb7F9) altered METH AUC and Cmax significantly which was accounted to 30 fold and 8 fold respectively	NCT03336866
IXT-m200 (ch-mAb7F9)	Humans		This phase II study is currently recruiting volunteers to investigate the effectiveness of IXT-m200 (ch-mAb7F9) in people with acute METH toxicity.	NCT04715230

KLH: Keyhole limpet hemocyanin, TT: Tetanus Toxoid, DT: Diphtheria Toxoid, BSA: Bovine serum albumin, mAb: Monoclonal antibody.

**Table 2 vaccines-10-01778-t002:** Summary of some selected preclinical and clinical studies on anti-cocaine immunotherapies.

Platform Description	Study Details	Model Tested	Ref
Catalytic antibodies	Three active catalytic mAbs identified by high-throughput assay procedure and evaluated for their ability to hydrolyze cocaine	In vitro	[50]
Catalytic antibodies	Six cocaine and one non-cocaine novel transition state analogs were synthesized, characterized and evaluated in animal models. 6 out 7 analogues demonstrated high anti-cocaine titers in mice	Mice	[49]
Catalytic antibodies	Catalytic antibody mAb 15A10 was produced using a transition-state analog for the hydrolysis of cocaine. mAb 15A10 was effective in protecting the rats in cocaine overdose model.	Rats	[52]
Viral gene transfer of cocaine hydrolage (CocH)	CocH at 0.3 or 1 mg/kg was effective in reducing drug levels in plasma and brain of mice given cocaine (10 mg/kg, subcutaneously or 20 mg/kg intraperitoneally). MAb at 8 mg/kg had little effect on cocaine distribution. CocH and mAb alone were not effective to suppress locomotor activity induced by high dose cocaine (100 mg/kg body weight) but these two candidates completely suppressed the locomotor activity when given in combination.	Mice	[62]
Hapten design and conjugation with carrier protein	Three fluorine-containing cocaine haptens (GNF, GNCF and GN5F) and one chlorine-containing cocaine hapten (GNCl) were synthesized, based on a chemical scaffold of succinyl norcocaine (SNC). These haptens were conjugated with KLH and evaluated in a mice model. GNF-KLH demonstrated higher affinity and antibodies compared to parent compound SNC	Swiss Webster mice	[54]
Hapten design and conjugation with carrier protein	Rats vaccinated with GNC-KLH did not restore cocaine self-administration behavior when given a non-contingent cocaine infusion for 2 days. Active immunization with GNC-KLH produced an 8-fold rightward shift of the dose-effect function for cocaine.	Rats	[63]
Catalytic and non-catalytic hapten design	The effectiveness of noncatalytic and catalytic anti-cocaine vaccine was evaluated in mice. A cocaine-like hapten GNE and a cocaine transition-state analogue GNT were conjugated with KLH and both vaccines demonstrated high levels of cocaine-specific antibodies and suppressed cocaine-induced locomotor behavior. However, with repeated cocaine administration antibodies and protecting effects of catalytic vaccine waned.	Mice	[55]
Anti-cocaine mAb	The effectiveness of the anti-cocaine mAb GNC92H2 was examined in a cocaine overdose model. 93 mg/kg (LD50) of cocaine was administered to Swiss albino mice. GNC92H2 mAb was administered at dose levels ranging from 30 to 190 mg/kg. Significant blockade of cocaine toxicity was observed with the higher dose of GNC92H2 (190 mg/kg).	Swiss albino mice	[64]
CocH	This study reported that CocH gene transfer therapy was effective to prevent the negative impact on the heart, brain reward system, and locomotor activity caused by cocaine metabolites.	Rats	[57]
CocH-Fc	A novel cocaine hydrolase catalyzing enzyme accelerated the metabolism of cocaine in rat blood even after 20 days of a single dose of CocH-Fc. This new construct has extended biological half-life (107 h) compared to the original enzyme (8 h)	Rats	[57]
Double mutant cocaine esterage (DM CocE)	A single dose of DM CocE was effective in catalyzing the cocaine in plasma. DM CocE effectively prevented the cocaine-induced increase in blood pressure, heart rate and locomotor activity.	rhesus monkeys	[59]
Double mutant cocaine esterage (DM CocE)	The repeated administration of DM CocH was effective in hydrolyzing the cocaine in the plasma within 5 to 8 min. The repeat administration of DM CocH produced anti-CocE antibodies, but it did not alter the effectiveness of DM CocE in metabolizing the cocaine in plasma, cardiovascular effects	rhesus monkeys	[60]
DM CocE	This study investigated the effectiveness of DM CocE against cocaine toxicity and reverse the cardiovascular toxicities. This study concluded that DM CocE was effective in protecting cocaine induced convulsion, cardiovascular changes and shifted the cocaine induced lethality to 10-fold right.	Rats	[65]
Active cocaine vaccine (TA–CD)	This was a double blind, randomized multicenter trial to evaluate the effectiveness of anti-cocaine vaccine (TA-CD) in 300 participants. In this study an IgG levels ≥ 42 μg/mL (high IgG) was satisfactory, and this level was achieved by 67% of the vaccinated participants receiving five vaccinations. Although for the full 16 weeks cocaine positive urine rates showed no significant difference among the three groups (placebo, high, low IgG), after week 8, more vaccinated than placebo subjects attained abstinence for at least two weeks of the trial (24% vs. 18%), and the high IgG group had the most cocaine-free urines for the last 2 weeks of treatment. However, neither was significant.	Clinical trial, phase III	[66]
Active cocaine vaccine (TA–CD)	This is a randomized, double blind phase II study conducted with 15 participants for a period of 13 weeks. TA-CD was administered at two dose levels (82 µg, *n* = 4; 360 µg, *n* = 6) at weeks 1, 3, 5, and 9. The level of antibody varied among the subjects and individuals with higher antibodies had immediate (within 4 min of cocaine smoking) and robust (55–81%) reduction in ratings of good drug effect and cocaine quality, while those in the lower half showed only a non-significant attenuation (6–26%).	Clinical trial, Phase II	NCT00965263 [67]

**Table 3 vaccines-10-01778-t003:** Clinical studies of anti-nicotine vaccines.

Clinical Studies				
Candidate	Hapten System	Carrier	Study Outcomes	Clinical Trial Identifier
NicVax	3′aminomethylnicotine	Pseudomonas aeruginosa rEPA	Phase III efficacy not demonstrated; increased abstinence related to antibody titer in phase II proof-of-concept	NCT01304810 NCT00598325 NCT00218413 NCT00318383 NCT00836199 NCT01102114
Nic002	O-succinyl-3′- hydroxymethylnicotine	VLP from bacteriophage Qβ	Primary end point not met in interim analysis of phase Iib study; per-protocol analysis of phase II study showed continuous abstinence rate at month 6 was 56% in high antibody group vs. 32.1% for placebo	NCT01280968 NCT00736047 NCT00369616
Niccine	IP18	Tetanus toxoid	Non-relapse rate at 1 year 43.3% for Niccine group versus 51.1% for placebo (95% CI = −20.6% to 4.9%)a	EudraCT 2007– 003250-29
TA-NIC	Nicotine N1-butryic acid	rCTB	Failed to demonstrate efficacy in phase II proof of- concept	NCT00633321
NIC7-001	5-aminoethoxy-nicotine	CRM	Study results not reported	NCT01672645
SEL-068	Nicotine	Proprietary polymer-based nanoparticle technology containing TLR agonist.	This study was designed to assess the safety and tolerability of subcutaneous injection of vaccine SEL-068. Study result not reported	NCT01478893
NicA2	Nicotine catalytic enzyme	Isolated from Pseudomonas putida S16,	This is study investigated the effect of NicA2 catalyzing enzymes. A NicA2 dose of 5 mg/kg reduced the brain Nic concentration 55% after 1 min and 92 % after Nic dose. The blood Nic concentration was below detection limit after the 1st or 5th Nic dose.	[75]
NicA-J1	Nicotine catalytic enzyme (reengineered)	Originally isolated from Pseudomonas putida S16 and then genetically engineered	This candidate completely prevented nicotine entry into rats’ brain and nicotine like compulsive behavior	[76]
NicA2 variants	Catalytic enzyme	NA	The investigators isolated and identified several NicA2 variants with improved efficacy. Among all the variants characterized, A107R reduced the nicotine entry to brain 3-fold higher than wild type and the PEGylation of the enzymes improved it shelf life in the circulation	[77]

**Table 4 vaccines-10-01778-t004:** Summary of selected preclinical and clinical studies on anti-opioid vaccines and mAbs.

Candidate Name	Composition	Adjuvants	Study Outcomes	Ref.
Morphine-BSA	morphine-6-hemisuccinate conjugated with bovine serum albumin (BSA)	Not reported	The rabbits immunised with the Morphine-BSA vaccine significantly alter the morphine clearance during the first four hours of the morphine injection (6 mg/kg BW).	[109]
M-6-S-BSA	A morphine-6-succinyl conjugated to immunogenic protein carrier BSA	Freund’s complete adjuvant	In this study, the developed vaccine was given to goats, rabbits, mice, and rats at the dose of 2 mg/kg BW. The vaccine was given weekly up to 7 weeks and on week 8, each animal was injected with 2 mg Morphine sulfate/kg BW to assess the efficacy. The vaccinated animal demonstrated reduced locomotor activity compared to the control group	[111]
M-6-S-BSA	A morphine-6-succinyl conjugated to immunogenic protein carrier BSA	Not reported	BALB/c mice and SD rats were treated with the vaccine and demonstrated strong (up to 1:200,000 and over 1:20,000) and morphine-specific antibody titers. Radiant heat tail-flick reflex test also demonstrated that this vaccine can reduce the antinociceptive against morphine	[112]
M-6-S-BSA	A morphine-6-succinyl conjugated to immunogenic protein carrier BSA	Not reported	347 morphine addicted people were vaccinated with M-6-S-BSA. Antibody response and safety was monitored one year. The antibody titre was at peak after the 3 months of the first injection and vaccine was well tolerated by the addicts.	[113]
M-TT	Morphine sulfate was conjugated with TT. A long spacer linker was used to connect the Morphine and TT	Not reported	This vaccine generated strong antibody response and prevented self-administration of heroin in immunized rats.	[114]
MorHap-TT Furthermore, combination of MorHap-TT+ palm-CV2 (HIV vaccine)	Morphine hapten was conjugated to carrier protein TT	MPL A ALF	In this study heroin and HIV vaccine was combined to assess the dual immunogenic profile. Immunised mice with both injections demonstrated satisfactory results. Palm-CV2induced anti-cyclic peptide titers at the degree of >106 and antibodies also prevented the binding of V2 peptide to the HIV-1 α4β7 integrin receptor. The anti-MorHap antibody was effective to prevent hyperlocomotion and antinociception induced by heroin.	[116]
6-AmHap-TT	A novel hapten 6-AmHap was synthesized and conjugated with protein carrier TT	Liposomal MPLA (ALFA)	This study reported that the novel vaccine generated strong antibody response against heroin in mice model and demonstrated cross reactivity with codeine, oxycodone, hydrocodone, hydromorphone, and oxymorphone	[117]
MorHap-TT and cross-reactive material 197 (CRM197).	Heroin/morphine hapten (MorHap) conjugated with TT and CRM197	L(MPLA)	Immunization of mice with these vaccines produced strong antibody titers (400–1500 ug/mL) against heroin and its metabolites 6-acetylmorphien and morphine. TT based vaccine demonstrated better inhibition of heroin induced antinociception which correlates with its hapten density.	[118]
M-KLH	Morphine is conjugated with KLH	Not reported	The study reported the development of M-KLH conjugate and assessment of of efficacy in rat models. The conjugated vaccine demonstrated strong antibody response and was able to attenuate heroin induces locomotor activity. The dopamine concentration in the brain was significantly lower in vaccinated mice compared to KLH group (126.08 ± 22.05 ng/mL vs. 45.58 ± 8.36 ng/mL)	[120]
Heroin/Morphine-KLH	Two heroin and morphine like haptens were synthesized and conjugated with KLH	Not reported	Heroin like vaccine system was effective to block self-administered of Heroin and antinociception induced by heroin.	[121]
KLH-6-SM	6-SM hapten was conjugated with KLH	Not reported	This study was designed to investigate the efficacy of a morphine like vaccine against morphine and other heroin like metabolites. This study reported that antibody binding was prevented by free morphine and heroin like metabolites, reduced antinociception caused by morphine and reduced the morphine concentration in the rat’s brain.	[122]
Heroin-KLH	Heroin in conjugated with immunogenic protein carrier KLH	TLR9 agonist CpG ODN 1826	The routes of immunization have been investigated by vaccinating the mice via SC and IP. Mice vaccinated via SC demonstrated inferior antibody responses compared to IP. CpG ODN 1826 increased the antibody response significanatly compared to control.	[123]
Various	Various hapten system has been conjugated with various carrier protein such as TT, DT, KLH	Al(OH)3 CpG ODN 1826	The study inbvestigated a series of hapten system and carrier protein along with adjuvants. A combination of hapten (HerCOOH), adjuvant (CpG ODN + alum), carrier protein (TT) was found to be efficacious.	[124]
Various	20 vaccine formulations have been investigated	TLR9 & TLR3 agonist	This study investigated the 20-vaccine formulation varying the carrier protein and adjuvants. TLR3 and TLR9 based vaccine formulation alone demonstrated strong antibody titer but combination of these two did not improve the antibody titre. Stability study revealed that TLR3 based formulation was more stable than TLR9. TLR9 + alum heroin vaccine formulation was effective to prevent the heroin lethal dose toxicities.	[125]
Heroin-TT/CRM and fentanyl-TT/CRM vaccine	Heroin and fentanyl is conjugated with TT/CRM along with adjuvants alum and CpG ODN	Advax/CpG ODN/δ-inulin	This study reported that inulin-based Heroin vaccine along with CpG ODN provided superior efficacy compared to other combination. Freeze dried vaccine formulation demonstrated stability up to one year at room temperature.	[126]
FEN-TT	FEN conjugated to TT	Liposome with MPLA adsorb on Alum	The investigators developed a liposomal conjugated vaccine system using MPL A and alum and reported that this vaccine demonstrated a strong antibody response in order of greater than 106 and the antinociceptive dose–response curve was shifted to the right	[132]
FEN-TT	FEN conjugated to TT	Not reported	This study investigated the effectiveness of the FEN-TT conjugated vaccine to alter the FEN self-administration in an experimental model called “fentanyl vs. food choice model”. This vaccine was effective to reduce the FEN reinforcement significantly and increased the food reinforcement. The study also demonstrated that this conjugated vaccine prevented the FEN withdrawal following 12 h FEN session.	[131]
FEN-sKLH and FEN-KLH	Fentanyl (FEN) conjugated to subunit KLH (sKLH) or KLH	Not reported	The study demonstrated that both FEN-KLH and FEN-sKLH reduced the hot plate-induced antinociception and distribution of Fentanyl to the brain. However, FEN-sKLH was more effective in reducing respiratory depression and overdose toxicity after cumulative administration of 50 µg/kg fentanyl dose.	[130]
FEN-BSA or FEN-TT	Fentanyl conjugated with BSA or TT	alum, dmLT, or LTA1	This study investigated the effect of various routes of administration and the adjuvant system. This is study demonstrated that FEN-TT conjugate with dmLT, or LTA1 adjuvant demonstrated superior efficacy when administered sublingually.	[127]
OXY-dKLH	Oxycodon conjugated with KLH dimer and adsorbed into	Alhydrogel	This study conducted in animal model confirmed that vaccinated mice demonstrated reduced effect on two dangerous cause of Oxycodone overdose fatality which are respiratory degression and heart rate.	[135]
(6OXY(Gly)4–KLH)	Oxycodone conjugated with tetra glycine linker and KLH immunogenic carrier	Not reported	The study reported that (6OXY(Gly)4–KLH) vaccine increased drug serum binding and reduced the distribution drug to the brain.	[134]
OXY-KLH	Oxycodone conjugated with KLH	Not reported	The study confirmed that OXY-KLH vaccine demonstrated increased amount of adenylate cyclase 5 (Adcy5), decreased amount of early growth response protein 2 (Egr2) and the early immediate gene c-Fos in the striatum. These findings further confirmed that this vaccine has the capability of reducing the reinforcing effects of oxycodone.	[133]
Oxy (Gly)4-sKLH	Oxycodone was linked with tetra glycine peptide linker and KLH immunogenic carrier	Not reported	A phase 1 & 2 study has been registered recently with clinicalTrial.gov (accessed on 11 October 2022) and currently recruiting participants. This trial is going to investigate the effect of vaccine Oxy (Gly)4-sKLH against oxycodone. This is a multisite study aiming to assess the safety, degree of antibody and efficacy.	NCT04458545

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
