# Peer review of "Immunotherapies for the Treatment of Drug Addiction"

_vaccines, 2022, doi:10.3390/vaccines10111778_

Round 1

Reviewer 1 Report

This article overviewed the current immunotherapies for the treatment of drug addiction, involved in mechanisms, platforms for vaccine development and immunotherapies against to various drug addiction disease. Generally, this paper provided updated research data and summary and could provide reference for related research. Only some minor revisions need to be addressed as follow.

Some minor revisions:

1. The number of keywords is generally no more than 5.

2. “Substance use disorders” is abbreviated as SUD in the beginning, but there are still a lot of full names in the text.

3. in Figure 2. Subfigure should be marked as a and b instead of Fig 2a and Fig 2b.

Author Response

Comments and Suggestions for Authors

This article overviewed the current immunotherapies for the treatment of drug addiction, involved in mechanisms, platforms for vaccine development and immunotherapies against to various drug addiction disease. Generally, this paper provided updated research data and summary and could provide reference for related research. Only some minor revisions need to be addressed as follow.

Author’s Feedback: The authors would like to appreciate your time to review this article. The authors have addressed all of your comments and believe that the overall quality of the manuscript has been improved.

  1. The number of keywords is generally no more than 5.

Author’s response: We have reduced the number to 5.

  1. “Substance use disorders” is abbreviated as SUD in the beginning, but there are still a lot of full names in the text.

Author’s response: We have replaced the full names by abbreviated form.

  1. in Figure 2. Subfigure should be marked as A and B instead of Fig 2a and Fig 2b.

Author’s response:  The figure sub-number has been adjusted as advised

Reviewer 2 Report

The Review by Hossain et al. presents summarized the development of immunotherapies for treating drug addiction. This Review is well-shaped and comprehensive, and several points of opinions are insightful. Considering the serious and urgent health and economic lost caused by SUD, this Review is very timely, and I hope that it will serve as an invaluable source for researchers and policy makers in the relevant fields. Therefore, we recommend the publication of this paper after addressing some minor concerns as listed below.

1) It is recommended that the authors can present the chemical structures for the drugs mentioned in this review as well as their relevant metabolites, and introduced modifications or sites of modifications for hapten design.

2) In Fig. 2b, “melanoma” should be “myeloma”. Similar mistake is in lines 123-124.

3) Because hapten designs and modifications are very important for developing immunotherapies for these small molecule drugs, probably more details should be provided for this research direction in section 3.1, or some relevant review papers can be cited here. Similar discussions on carrier proteins are also expected.

4) The role of adjuvant is not discussed. Probably the authors may want to cite a review paper on this subject (Adjuvants for vaccines to drugs of abuse and addiction).

5) There is a previous PNAS paper studying potential LSD antibody (10.1073/pnas.68.7.1483

). I think this paper may also be of interest for this review. Further, it is recommended to include some discussions on some so-called “novel drugs of abuse” that are not included in this review and if there is any immunotherapy effort targeting them.

6) In lines 169-170, the implication for an induced increase in AUC and Cmax should elaborated.

7) The title for section 10 is “Conclusion and five-year view”, but I could not find any information provided on “five-year view” specifically.

8) Some gramma mistakes or typos, just listed a few as below. More carefully proofreading should be conducted.

Line 46: “current treatments are currently lacking”, current and currently are repeated.

Line 183: “hydrolyze the compound render the, ineffective” is not a complete sentence.

Line 194: “a novel CocH through fusing with a catalytic antibody fragment crystallizable (Fc)” should be “a novel CocH through fusing with an antibody fragment crystallizable (Fc)”.

Line 358: “This study also reported that PEGylation of the enzymes also improved the circulation half-life”, two “also” are repeated.

Lines 446-447: “In a another study conducted by”.

Line 522: “have provided critical information and based on which scientists are working”, “and” should be deleted.

Author Response

Comments and Suggestions for Authors

The Review by Hossain et al. presents summarized the development of immunotherapies for treating drug addiction. This Review is well-shaped and comprehensive, and several points of opinions are insightful. Considering the serious and urgent health and economic lost caused by SUD, this Review is very timely, and I hope that it will serve as an invaluable source for researchers and policy makers in the relevant fields. Therefore, we recommend the publication of this paper after addressing some minor concerns as listed below.

Author’s Feedback: The authors would like to appreciate your time to review this article. The authors have addressed all of your comments and believe that the overall quality of the manuscript has been improved.

  • It is recommended that the authors can present the chemical structures for the drugs mentioned in this review as well as their relevant metabolites, and introduced modifications or sites of modifications for hapten design.

Author’s Response: Thank you for your suggestion. Recently the following article has been published covering the hapten structure and is very insightful. This has been referenced in the revised paper.

https://pubs.rsc.org/en/content/articlelanding/2021/cb/d0cb00165a

However, the objective of our article is to provide a cursory overview and provide a general update on the most commonly abused drugs. The scope of the article is already broad and hence we are taking your suggestion on board and considering this as our future project.

2) In Fig. 2b, “melanoma” should be “myeloma”. Similar mistake is in lines 123-124.

 Author’s response: The suggested spelling error has been corrected.

3) Because hapten designs and modifications are very important for developing immunotherapies for these small molecule drugs, probably more details should be provided for this research direction in section 3.1, or some relevant review papers can be cited here. Similar discussions on carrier proteins are also expected.

 Author’s response: The authors expanded the discussion as advised with citations of relevant papers.

4) The role of adjuvant is not discussed. Probably the authors may want to cite a review paper on this subject (Adjuvants for vaccines to drugs of abuse and addiction).

Author’s response:  The authors have cited the reference review article in the text (section 3.1).

5) There is a previous PNAS paper studying potential LSD antibody (10.1073/pnas.68.7.1483). I think this paper may also be of interest for this review. Further, it is recommended to include some discussions on some so-called “novel drugs of abuse” that are not included in this review and if there is any immunotherapy effort targeting them.

Author’s response: A brief discussion on “novel drugs of abuse” has been included

6) In lines 169-170, the implication for an induced increase in AUC and Cmax should be elaborated.

Author’s response: Edited as suggested.

7) The title for section 10 is “Conclusion and five-year view”, but I could not find any information provided on “five-year view” specifically.

Author’s response: The authors have removed the word “ five year view”

8) Some grammar mistakes or typos, just listed a few as below. More carefully proofreading should be conducted.

Author’s response: All the authors including a few native English speakers have reviewed the article to improve the grammar and avoid typos. All the suggested spelling errors and typos have been corrected

Line 46: “current treatments are currently lacking”, current and currently are repeated.

Author’s response: The typo has been corrected

Line 183: “hydrolyze the compound render the, ineffective” is not a complete sentence.

Author’s response: The sentence is complete now

Line 194: “a novel CocH through fusing with a catalytic antibody fragment crystallizable (Fc)” should be “a novel CocH through fusing with an antibody fragment crystallizable (Fc)”.

Author’s response: The sentence has been modified as advised

Line 358: “This study also reported that PEGylation of the enzymes also improved the circulation half-life”, two “also” are repeated.

Author’s response: The additional “also” has been removed

Lines 446-447: “In a another study conducted by”.

Author’s response: The grammar has been corrected

Line 522: “have provided critical information and based on which scientists are working”, “and” should be deleted.

Author’s response: The additional “and” has been removed.

Reviewer 3 Report

This manuscript by Hossain et al. summarized various immunotherapies for the drug-induced addiction. Overall, it’s a well-written and interesting review. To strengthen the manuscript and broaden the readership, I would recommend the following revisions:

1.     This manuscript makes many claims of fact without supporting citations of evidence - e.g., line 49-56.

2.     In the introduction, you can split the first paragraph into two, including epidemiology and current therapy.

3.     The manuscript requires editorial assistance, focusing on both grammars as well the overall structure of the manuscript such as the ordering of the paragraphs, paragraph structure and sentence structure, such as line 57-59.

4.     You should provide more details of figure legends and tables.

5.     Could you please add more summary at the start of each part and make it flow naturally? And you need to state the conclusions at the end of different part.

6.     Part 4-7 are the immunotherapies for different substance abuse disorders. I think you can modify your titles.

Author Response

Comments and Suggestions for Authors

This manuscript by Hossain et al. summarized various immunotherapies for the drug-induced addiction. Overall, it’s a well-written and interesting review. To strengthen the manuscript and broaden the readership, I would recommend the following revisions:

  1. This manuscript makes many claims of fact without supporting citations of evidence - e.g., line 49-56.

Author’s response: Some supportive references have been cited.

  1. In the introduction, you can split the first paragraph into two, including epidemiology and current therapy.

Author’s response: The introduction section has been split into two paragraphs.

  1. The manuscript requires editorial assistance, focusing on both grammars as well the overall structure of the manuscript such as the ordering of the paragraphs, paragraph structure and sentence structure, such as line 57-59.

Author’s response: The article has been reviewed by a native English speaker to improve the grammar and typos.

  1. You should provide more details of figure legends and tables.

Author’s response: There are further explanations of each figure and tables in the respective sections of the manuscript.

  1. Could you please add more summary at the start of each part and make it flow naturally? And you need to state the conclusions at the end of different part.

Author’s response: Thank you for your comment, we flow-on sentences have been included

  1. Part 4-7 are the immunotherapies for different substance abuse disorders. I think you can modify your titles.

Author’s response: Thank you for your comment, edited as requested